# Prodigiosin of *Serratia marcescens* ZPG19 Alters the Gut Microbiota Composition of Kunming Mice

**DOI:** 10.3390/molecules26082156

**Published:** 2021-04-09

**Authors:** Xue Li, Xinfeng Tan, Qingshuang Chen, Xiaoling Zhu, Jing Zhang, Jie Zhang, Baolei Jia

**Affiliations:** 1State Key Laboratory of Biobased Material and Green Papermaking, School of Bioengineering, Qilu University of Technology (Shandong Academy of Sciences), Jinan 250000, China; 1043118284@stu.qlu.edu.cn (X.L.); T1664294917@163.com (X.T.); qingshuangchen95@163.com (Q.C.); 2Shandong Academy of Agricultural Sciences, Jinan 250000, China; zhuxl2001@126.com

**Keywords:** gut microbiota, Kunming mice, pathologic visceral changes, prodigiosin, *Serratia marcescens*

## Abstract

Prodigiosin is a red pigment produced by *Serratia marcescens* with anticancer, antimalarial, and antibacterial effects. In this study, we extracted and identified a red pigment from a culture of *S. marcescens* strain ZPG19 and investigated its effect on the growth performance and intestinal microbiota of Kunming mice. High-performance liquid chromatography/mass spectrometry revealed that the pigment had a mass-to-charge ratio (*m*/*z*) of 324.2160, and thus it was identified as prodigiosin. To investigate the effect of prodigiosin on the intestinal microbiota, mice (n = 5) were administered 150 μg/kg/d prodigiosin (crude extract, 95% purity) via the drinking water for 18 days. Administration of prodigiosin did not cause toxicity in mice. High-throughput sequencing analysis revealed that prodigiosin altered the cecum microbiota abundance and diversity; the relative abundance of *Desulfovibrio* significantly decreased, whereas *Lactobacillus reuteri* significantly increased. This finding indicates that oral administration of prodigiosin has a beneficial effect on the intestinal microbiota of mice. As prodigiosin is non-toxic to mouse internal organs and improves the mouse intestinal microbiota, we suggest that it is a promising candidate drug to treat intestinal inflammation.

## 1. Introduction

Prodigiosin is a red microbial pigment with a tripyrrole ring structure [1,2,3]. It is mainly produced by *Serratia marcescens* [4,5], *Pseudomonas*, *Actinomycetes*, and some marine bacteria [6,7,8]. Prodigiosin appears red under acidic and neutral conditions and yellow under alkaline conditions [9,10]. As a bioactive substance, it has anticancer [11], antimalarial [12], antibacterial [13,14], antigenic [15,16], and immunosuppressive effects. Prodigiosin inhibits algae growth and is used to control natural water pollution, such as that caused by red tides and blooms [17]. Further, it is sensitive to light and is added to sunscreens and cosmetics to reduce UV damage to the skin [18]. Thus, prodigiosin has broad application prospects in the fields of medicine, environmental management, and cosmetics.

Prodigiosin can be classified into cycloprodigiosin, undecylprodigiosin, and metacycloprodigiosin (Figure 1). Cycloprodigiosin produced by the marine bacterium *Pseudoalteromonas denitrificans*, promotes inducible nitric oxide synthase gene expression in interleukin-1β-stimulated hepatocytes [19]. Undecylprodigiosin, produced by the marine actinomycete *Saccharopolyspora* sp. *novSP2-10*, is considered a common precursor of the prodigiosin family. It reduces microsclerotia formation by *Verticillium dahliae* in vitro [8]. Similar to undecylprodigiosin, metacycloprodigiosin selectively induces β-catenin-mutant tumor cell death [20]. Thus, prodigiosin exhibits species-specific functional differences.

Previous studies on prodigiosin have focused on the screening of high-yielding strains, optimization of the fermentation process, separation and purification, and in vitro biological activities; however, the effect of prodigiosin on the intestinal microbiota of mice has not been investigated. It has been shown that oral administration of lactate and pyruvate enhances the immune response and strengthens the intestinal resistance to *Salmonella* infection in mice [21]. Extracellular polysaccharides, a metabolite of Bifidobacterium, significantly increased the growth of *Lactobacillus* and anaerobic bacteria and inhibited *Enterobacter*, *Enterococcus* and *Bacteroides fragilis* [22]. All these studies suggest that microbial metabolites can enhance intestinal microecological balance. As the intestinal microbiota is closely related to health, and metabolites are the material basis for the communication between the intestinal flora and the host [23], we investigated the effect of oral administration of prodigiosin on health in mice. To this end, we screened a strain of *S. marcescens* and purified and identified its pigment. Crude prodigiosin extract was used to study its effects on mouse health, and histopathological observation and intestinal microbiological analysis were used to provide a scientific basis for the safety of orally administered prodigiosin.

## 2. Materials and Methods

### 2.1. Isolation of S. marcescens

*S. marcescens* was isolated from compost generated by aerobic composting of *Flammulina velutipes* residue that was anaerobically fermented for biogas production (Dezhou, China) and was stored at 25 °C. One gram of compost was immersed in physiological saline for 2 h. After serially diluting the sample solution, 200 μL was added onto Luria–Bertani (LB) agar containing 0.075 g/L K_2_Cr_2_O_7_ for isolation. The medium’s pH was adjusted to 7.2 with NaOH solution, and then the medium was autoclaved. The plates were incubated at 30 °C. Red colonies were picked and continuously streaked to obtain a pure strain. The strain was named *S. marcescens* ZPG19 and was deposited in the China Centre for Type Culture Collection (CCTCC M 2019645). Colony morphology was observed under a microscope (Eclipse E200; Nikon, Tokyo, Japan), and cell morphology was observed under a scanning electron microscope (SUPRA™ 55; Carl Zeiss, Jena, Germany).

### 2.2. 16S rRNA Gene-based Phylogenetic Analysis

Genomic DNA of ZPG19 was extracted from cell biomass cultivated in LB broth. The nearly full-length 16S rRNA gene was amplified using 27F (5′-AGAGTTTGATCCTGGCTCAG-3′) and 1492R (5′-CTACGGCTACCTTGTTACGA-3′) primers. The reaction mixture composition and PCR conditions were as previously reported [24]. The amplified 16S rRNA gene fragment was sequenced (Personal Biotechnology, Shanghai, China), and the sequence was compared with homologous sequences available in GenBank using the NCBI BLAST program (http://www.ncbi.nlm.nih.gov/blast, accessed on 7 December 2020). The nucleotide sequence of the 16S rRNA gene of strain ZPG19 was deposited in GenBank under accession number MW308502. Phylogenetic investigation of communities by reconstruction of unobserved states (PICRUSt) [25] was used to compare the 16S rRNA gene sequence data with the Kyoto Encyclopedia of Genes and Genomes (KEGG) database to compare the abundances of functional genes in biological metabolic pathways.

### 2.3. Extraction and Identification of the Red Pigment

1 mL *S. marcescens* culture was centrifuged at 9000× *g* for 10 min under 4 °C. After aspirating the supernatant, 1 mL of acidic methanol (pH 2–3) was added for ultrasonic-assisted extraction for 30 min [26]. Then, the red pigment-acidic methanol solution was centrifuged at 9000× *g* for 10 min. The supernatant was first analyzed by high-performance liquid chromatography (HPLC). An SB-C18 reversed-phase column (ZORBAX, 4.6 × 150 mm^2^, 5 μm, Agilent, Shanghai, China) was used for isocratic elution with 85:15 (*v*/*v*) acetonitrile:ammonium acetate (5% aqueous solution) for 10 min. The flow rate was 1 mL/min at room temperature. The injection volume was 10 μL. The mass of redox was then determined by high-performance liquid chromatography/mass spectrometry (HPLC-MS) using a 1200 RRLC-6410 Triple Quad LC/MS instrument (Agilent, Santa Clara, CA, USA) under an ESI (+) mode, with a mass scan range of 50–600 [27].

### 2.4. Effect of Prodigiosin on Mice

All animal experiments were performed according to the protocols approved by the Institutional Animal Care and Use Committee of the Qilu University of Technology (the approved protocol number is #201904116). Healthy Kunming mice (weight, 28 ± 2 g; age, 5 weeks) were purchased from Jinan Pengyue Laboratory Animal Breeding (Experimental Animal Use License: SCXK (Lu) 20140007). The mice were housed under natural ventilation, with free access to food and drinking water for a 37-day laboratory adaptation period. The litter was replaced regularly.

The prodigiosin used for mouse feeding was extracted from the fermentation broth by ultrasonic-assisted methanol extraction. Briefly, 1 mL of solution was analyzed for the concentration and purity of prodigiosin. After removing the methanol by rotary evaporation, the prodigiosin was placed in a fume hood and allowed to stand for some time to obtain solid prodigiosin. 42 mg prodigiosin (purity 95% HPLC, peak area normalization method) was weighed and pre-dissolved in 1 mL of absolute ethanol to make it soluble in water.

The mice were divided into two groups (n = 5) and fed prodigiosin at a dose of 0 μg/kg/d or 150 μg/kg/d for the control group (C) and prodigiosin group (PF), respectively. As 100 μg/kg/d prodigiosin was used to treat brain injury by intravenous injection [28], a prodigiosin dose of 150 μg/kg/d oral administration was used in this study. During the experiment, mice in groups C and PF were fed the same diet (Appendix A). For both groups, anhydrous ethanol and a 40 µg/mL prodigiosin-ethanol solution were dissolved in water, and the dose was continuously adjusted according to the amount of water consumed to ensure that mice received a daily dose of 150 µg/kg/d and were monitored once daily.

After the experiment, three mice with similar weight and in good condition were selected from each treatment group. The mice were killed by cervical dislocation, and various organs and the intestines were collected. Heart, liver, spleen, lung, and kidney samples were stained with hematoxylin–eosin and photographed under a microscope. Cecum contents were collected for high-throughput sequencing.

### 2.5. Microbiota Analysis

Total bacterial DNA was extracted from cecum contents of three randomly selected mice in each group using the HiPure stool DNA kit (Magen, Guangzhou, China) and was used as a template to amplify the V3-V4 variable regions of the 16S ribosomal (r)RNA gene with specific primers (341F: 5′-CCTACGGGNGGCWGCAG-3′, 806R: 5′-GGACTACHVGGGTATCTAAT-3′). Amplicons were extracted from 2% agarose gels and purified using the AxyPrep DNA gel extraction kit (Axygen Biosciences, Union City, CA, USA) according to the manufacturer’s instructions and quantified using an ABI StepOnePlus real-time PCR system (Life Technologies, Foster City, CA, USA). Purified amplicons were pooled in equimolar concentrations and paired-end sequenced (PE250) on the Illumina platform (Gidio Biotechnology, Guangzhou, China) according to the standard protocols. Raw sequence data were quality-screened and assembled using Flash software [29]. UCHIME was used to identify interrogative sequences and eliminate chimeras and count high-quality sequences [30]. Clean tags were clustered into operational taxonomic units (OTUs) according to a 97% similarity level using UPARSE (version 9.2.64). A Venn diagram was prepared to show common and unique OTUs among samples [31]. Abundance and diversity indices were calculated and presented using R QIIME [32].

### 2.6. Statistical Analysis

Data were analyzed and processed using SPSS 18.0. Data are presented as the mean ± standard deviation (SD). *p* < 0.05 was considered significant. Linear discriminant analysis effect size (LEfSe) analysis was used to analyze differences between the groups. LEfSe first applies the Kruskal–Wallis rank-sum test to assess differences in gene abundance between all groups and then uses the Wilcoxon rank-sum test to compare pairs of groups. Finally, significant differences were sorted based on linear discriminant analysis (LDA) scores.

## 3. Results

### 3.1. Phenotypic and Phylogenetic Analysis and Identification of ZPG19

Strain ZPG19 was found to be an aerobic, Gram-negative bacterium. After a 24 h cultivation on LB agar, the red pigment was produced. Colonies were round, bulged in the middle and with regular edges, and appeared red and opaque (Figure 2a). Light microscopy and scanning electron microscopy revealed that cells were short and rod-shaped, with a length of approximately 1.6 μm (Figure 2b,c).

Comparison of the 16S rRNA sequence of strain ZPG19 with closely related taxa retrieved from GenBank confirmed that the strain belongs to the genus *Serratia* (phylum Proteobacteria). In phylogenetic analysis, the highest sequence similarity (99%) was observed between ZPG19 and the type strain *S. marcescens* ss04 (Figure 2d). These findings suggested that ZPG19 is a strain of *S. marcescens*, and thus we named it *S. marcescens* ZPG19.

### 3.2. Identification of Metabolites of S. marcescens ZPG19

The red pigment from *S. marcescens* ZPG19 was extracted and analyzed by HPLC (Figure 3a) and HPLC-MS. HPLC-MS (Figure 3c) revealed that the mass-to-charge ratio (m/z) of the substance corresponding to the peak was 324.2160, and its molecular weight was 323, which is the molecular weight of prodigiosin [33]. We preliminarily concluded that the red pigment produced by *S. marcescens* ZPG19 was prodigiosin, with a purity of 95.16% (Figure 3d). In the HPLC spectrum of the prodigiosin sample, there were two other small peaks (RT2 = 5.236, RT3 = 6.436) in addition to the main peak (RT1 = 4.356). HPLC-MS analysis revealed the following m/z values: RT1 = 324.2160, RT2 = 272.1995, and RT3 = 272.1937 (Figure 3c). In the infrared spectrum in Figure 3b, the signal at 3292 cm^−1^ represents an O–H bond, that at 2907 cm^–1^ a C–H bond, that at 1633 cm^–1^ an N–H bond, that at 1379 cm^–1^ a C–O bond stretching vibration or an O–H bond surface bending vibration, and that at 1091 cm^–1^ a C–O–C asymmetric stretching vibration. Based on the molecular weight and the fact that prodigiosin’s basic structure is a tripyrrole ring, plus that the structures of various prodigiosin homologs differ in the alkyl group on the C ring, the structures of RT2 and RT3 were predicted (Figure 3e,f). A more detailed analysis of the structures would require nuclear magnetic resonance spectroscopy after further separation and purification of the pigment.

### 3.3. Effect of Prodigiosin on the Internal Organs of Mice

To analyze the effect of prodigiosin on mouse health, mice were orally administered prodigiosin at 150 μg/kg/d for 18 days, and ethanol solution was used as a control. Bodyweight was similar between the two groups (Table 1). Microscopic observation of hearts collected from the mice administered 150 μg/kg/d prodigiosin for 18 days revealed that cardiomyocytes did not degenerate, myocardial fibers were arranged naturally and orderly, the muscle space was not significantly widened, and there was no inflammatory cell infiltration (Figure 4a). Liver cells were uniform in size, with no abnormalities in the nucleus and abundant cytoplasm (Figure 4b). Spleen lymphocytes did not decrease, and there was no common inflammation, such as red blood cell exudation (Figure 4c). Lung alveoli were uniform in size, with no inflammatory cell infiltration (Figure 4d). There were no abnormalities, such as hyperemia, in the renal interstitium, and there was no inflammatory cell infiltration (Figure 4e). These histopathological observations showed that 150 μg/kg/d prodigiosin for 18 days had no significant toxic effects on mouse viscera.

### 3.4. Effects of Prodigiosin on Mouse Gut Microbiota Diversity and Richness

We collected cecum samples after 18 days of prodigiosin feeding. Next-generation sequencing of the 16S rRNA genes was performed to analyze the microbiota composition. In total, 648,351 reads from six cecum samples were obtained, and 6417 OTUs were identified with a 97% similarity cutoff (Appendix A). The Venn diagram in Appendix A shows that the C group had 230 unique OTUs and the PF group had 506 unique OTUs, accounting for 56.57% and 74.13%, respectively. Thus, the total number of OTUs in the PF group was higher than that in the C group, indicating that prodigiosin increases gut microbiota community richness. The α-diversity indices of species diversity and richness for the two groups are shown in Appendix A. For both groups, the coverage was 0.99, which is close to 1, indicating that the results for both datasets reflect the real diversity. Appendix A shows that the Sobs (*p* = 0.27), Chao (*p* = 0.19), and ACE (*p* = 0.23) indices were slightly higher in the PF group than in the C group, indicating that the species abundance in the PF group tended to be higher than that in the C group. The Simpson and Shannon indices were roughly the same, indicating that the two groups had similar species diversity.

### 3.5. Prodigiosin Alters the Gut Microbiota Composition

The relative abundances of microbial taxa were compared between the two groups. The 10 most abundant microbial taxa at the phylum and genus levels are shown in Figure 5. At the phylum level (Figure 5a), Firmicutes, Bacteroides, and Proteobacteria were the dominant phyla in both the C and PF groups, accounting for 61.24%, 34.93%, 2.60% (total, 98.78%) and 52.77%, 44.60%, and 1.14% (total, 98.52%), respectively. Compared with the control treatment, prodigiosin administration increased the abundances of Bacteroidetes and Actinobacteria and decreased those of Firmicutes and Proteobacteria. At the genus level (Figure 5b), *Lachnospiraceae NK4A136*, *Prevotellaceae UCG-001*, *Lactobacillus*, *Desulfovibrio*, and *Eubacterium xylanophilum* were the dominant genera in the C group, accounting for 22.05%, 3.39%, 2.76%, 2.40%, and 1.48%, respectively. In the PF group, *Lachnospiraceae NK4A136*, *Alloprevotella*, *Ruminococcaceae UCG-014*, *Bacteroides*, and *Lactobacillus* were the dominant genera, accounting for 17.61%, 4.13%, 2.66%, 2.27%, and 1.89%, respectively. Compared with the control treatment, prodigiosin administration increased the abundances of *Ruminococcaceae UCG-014*, *Alloprevotella,* and *Bacteroides* and decreased those of *Desulfovibrio*, *Prevotellaceae_UCG-001*, and *Prevotellaceae_NK3B31_group.*

The LEfSe approach was applied to identify the key phylotypes responsible for the difference in intestinal microbiota between the C and PF groups. *Ruminococcus_1*, *Caproiciproducens*, *Anaerotruncus*, *ASF356,* and *Eubacterium_nodatum_group*, which were the most abundant in the C group, and *Methanobacterium*, *Ruminiclostridium_9*, *Proteiniphilum*, and *Lachnospiraceae_UCG_006*, which were the most abundant in the PF group, were the dominant genera that contributed most to the difference in intestinal microbiota between the two groups. Notably, *Lactobacillus reuteri* (*p* = 0.04) in the PF group and *Desulfovibrio fairfieldensis* (*p* = 0.04) in the C group were the dominant species contributing to the difference in intestinal microbiota between the C and PF groups (Figure 5c).

### 3.6. Effects of Prodigiosin on Intestinal Microbiota Functions in Mice

To predict the microbiota’s metabolic functions, PICRUSt analysis was conducted based on the KEGG metabolic pathways [34].

There were four functional categories related to cellular processes, in which cell motility was the most strongly enriched function in the C and PF groups, with 128,725 and 102,031 genes, respectively, followed by cell growth and death, with 47,530 and 41,409 genes, respectively. Transport and catabolism and cellular community-prokaryotes had similar gene contents. These four metabolism-related functional categories were more enriched in the C group than in the PF group.

There were three functional categories related to environmental information processing, and membrane transport-related genes were the most abundant in each sample, with 58,842 and 48,568 genes in the C and PF groups, respectively. There were no signaling genes or interaction-related genes in the C group.

There were four functional categories related to human diseases, of which infectious diseases were more enriched in functional genes in the C group than in the PF group. There were no neurodegenerative disease-related genes in the PF group. Interestingly, the infectious diseases and neurodegenerative diseases categories were more enriched in the C group (Figure 6).

## 4. Discussion

In this study, a red pigment-producing strain was isolated from compost generated by aerobic composting of *Flammulina velutipes* anaerobic ferment. Based on colony morphology and physiological and biochemical characteristics combined with 16S rRNA gene identification, the isolated strain ZPG19 was identified as *S. marcescens*. We used ultrasonic-assisted acid methanol extraction to extract prodigiosin from the bacteria, and HPLC was used to determine that the prodigiosin had a purity of 95.16%. Notably, in addition to the main prodigiosin peak, two unknown peaks were observed in the HPLC spectrum. Based on HPLC-MS results, it was speculated that they might represent prodigiosin homologs. There are many derivatives of prodigiosin. For example, undecylprodigiosin isolated from *Streptomyces* sp. JS520 has been used as a natural protective agent to alleviate cell plasma membrane damage, opening up prospects for the plasma medicine field to protect the human body from excessive exposure [35]. Cycloprodigiosin, isolated from the marine bacterium *P. denitrificans*, increases the production of NO through interleukin-1b, thereby exerting an antitumor effect [19]. Although prodigiosin family members have different sources and functions, their basic skeletons are all tripyrrole rings. Therefore, we conclude that the main secondary metabolites of ZPG19 are prodigiosin (>95%) and its homologs (<5%). Natural products are generally complex, and as these three compounds are homologs, their chemical properties are very similar, which makes it difficult to completely separate them with conventional purification methods. Therefore, further separation and identification will be required to fully elucidate the structures of the homologs.

Oral administration of 150 μg/kg/d prodigiosin for 18 days had no significant effect on the weight of mice and no obvious toxic effect on the internal organs. High-throughput sequencing results showed that prodigiosin altered the composition of the intestinal microbiota of mice, which may reduce the occurrence of enteritis in mice. At the phylum level, prodigiosin decreased the abundances of Firmicutes and Proteobacteria and increased those of Bacteroides and Actinomycetes. Many harmful bacteria belong to Firmicutes and Proteobacteria, e.g., *Listeria* and *Clostridium* belong to Firmicutes, and *Salmonella* and *Pseudomonas aeruginosa* are Proteobacteria. Numerous Proteobacteria species can produce endotoxins, which can cause mucosal structural and functional damage through direct or indirect interaction with the mucosa, thus increasing permeability, inducing immune changes, and mediating local or systemic inflammation [36,37,38]. Thus, the decrease in the abundances of Firmicutes and Proteobacteria indicates a positive effect of prodigiosin on the mouse intestine. Bacteroidetes help the host absorb nutrients [39,40], promote fat accumulation [41], and can induce expression to specifically kill Gram-positive bacteria [42]. A metabolomics study has shown that bacteria in the phylum Bacteroides can express various complex polysaccharide-degrading enzymes [43]. Further, a previous study identified and characterized bacteria that produced metabolites with antitumor effects [44]. Phylogenetic analysis showed that most of them belonged to the phylum Actinobacteria. Together, these and our findings indicate that prodigiosin benefits the intestinal microbiota of mice and may reduce the risk of malignant tumor development.

At the genus level, prodigiosin increased the abundances of *Alloprevotella*, *Ruminococcaceae UCG-014,* and *Bacteroides* and decreased those of *Desulfovibrio*, *Prevotellaceae_UCG-001*, and *Prevotellaceae_NK3B31_group*. Transplantation of fecal microbiota containing *Ruminococcaceae UCG-014* into dysplastic mice improved their growth and metabolic abnormalities and inhibited *Vibrio cholerae* infection [45]. *Alloprevotella* produces short-chain fatty acids and anti-inflammatory substances, and Xiexin Tang, a traditional Chinese medicine, improved type II diabetes in rats by increasing *Alloprevotella* abundance [46]. An increase in intestinal Bacteroides is negatively correlated with metabolic diseases, such as obesity and diabetes, and depression and metabolic disease [47]. An increase in intestinal *Desulfovibrio* is an important feature of polyps and ulcerative colitis, and the gut microbial diversity of human patients with intestinal diseases is lower than that of healthy subjects, the dominant microbiota is shifted, and the microbiota is imbalanced [48]. An increase in the abundance of *Desulfovibrio* with a decrease in that of *Clostridia* leads to metabolic diseases [49]. This implies that prodigiosin reduces the probability of intestinal inflammation in mice by decreasing the abundance of *Desulfovibrio*. Studies have shown that prodigiosin family compounds can selectively regulate the proliferation of T lymphocytes [50,51,52] and regulate the response of macrophages to inflammatory stimuli [53]. Prodigiosin has strong antibacterial activity against *Escherichia coli*, *Pseudomonas aeruginosa*, *Staphylococcus aureus*, *Enterococcus faecalis*, *Streptococcus pyogenes*, *Acinetobacter* sp., *and oxacillin-resistant S. aureus* [54]. Together, these findings confirm that prodigiosin has anti-inflammatory and immune-enhancing effects.

Piewngam et al. showed that the *Bacillus* capable of producing fengycin by gavage could completely eliminate the colonization of *S. aureus* in the intestine [55]. Dabour et al. showed that pediocin PA-1 could effectively inhibit the infection of *Listeria monocytogenes* without affecting the original structure of the intestinal flora [56]. Bauerl et al. showed that plantaricins EF produced from *Lactobacillus* can temporarily and effectively improve the structure of intestinal flora at the genus level [57]. These studies have shown that microbial metabolites have a profound impact on the microbial ecological balance in the body and even the health of the body. In this study, we showed that prodigiosin from *S. marcescens* altered gut bacterial community in mice: *Lactobacillus reuteri* was significantly increased, and *Desulfovibrio fairfieldensis* was significantly decreased in the PF group compared to the C group. *Lactobacillus reuteri* produces reuterin, which inhibits harmful fungi, bacteria, and protozoa. Therefore, *L. reuteri* is considered an effective therapeutic agent for alleviating gastrointestinal diseases [58]. *Desulfovibrio fairfieldensis* can cause bacteremia and abscesses in the liver and abdomen, pneumonia, and urinary system infections [59,60]. Therefore, we suggested that prodigiosin from *S. marcescens* is a beneficial molecule for host health through modulating gut bacterial community.

In conclusion, we isolated a strain of *S. marcescens* from the bacterial residue of a compost, extracted a red pigment from it, and identified its structure. The pigment was identified as prodigiosin and its homologs. In mice, crude prodigiosin extract significantly altered the relative abundances of bacteria in the cecum: harmful bacteria decreased, whereas beneficial bacteria increased. Our findings indicated that prodigiosin had a beneficial effect on the intestines of Kunming mice. As prodigiosin was found to be non-toxic to mouse internal organs and to improve the mouse intestinal microbiota, it enabled further studies to analyze its effect on intestinal diseases to be carried out.

## Figures and Tables

**Figure 1 molecules-26-02156-f001:**
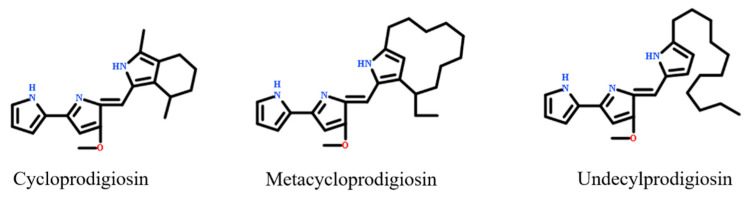
Structures of the cycloprodigiosin, undecylprodigiosin, and metacycloprodigiosin.

**Figure 2 molecules-26-02156-f002:**
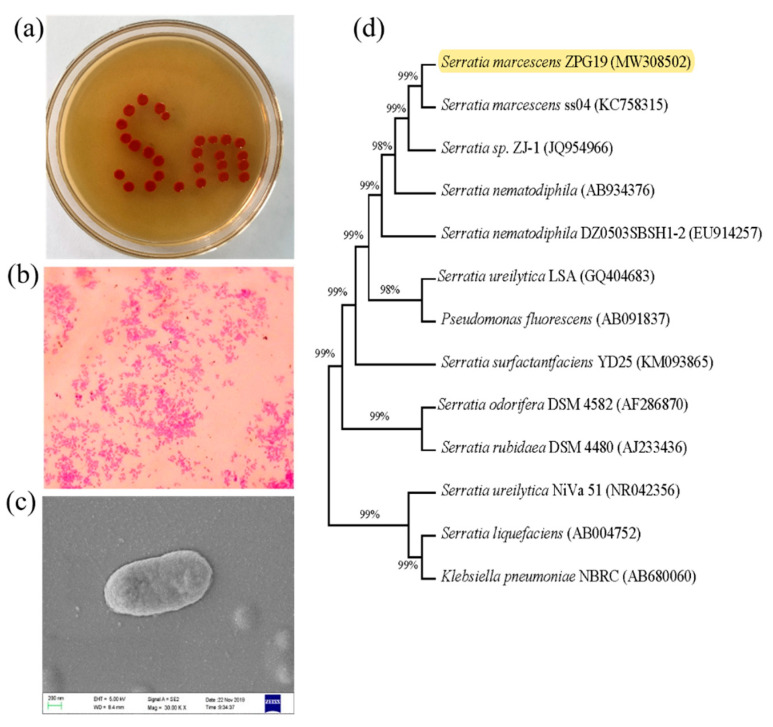
Phenotypic characteristics of *Serratia marcescens* ZPG19. (**a**) Colony morphology, (**b**) Gram staining, (**c**) scanning electron microphotograph (magnification, 300,000×). (**d**) Neighbor-joining phylogenetic tree based on 16S rRNA gene sequences of *S. marcescens* ZPG19 and closely related strains. The accession numbers of the sequences are indicated in brackets after the strain names.

**Figure 3 molecules-26-02156-f003:**
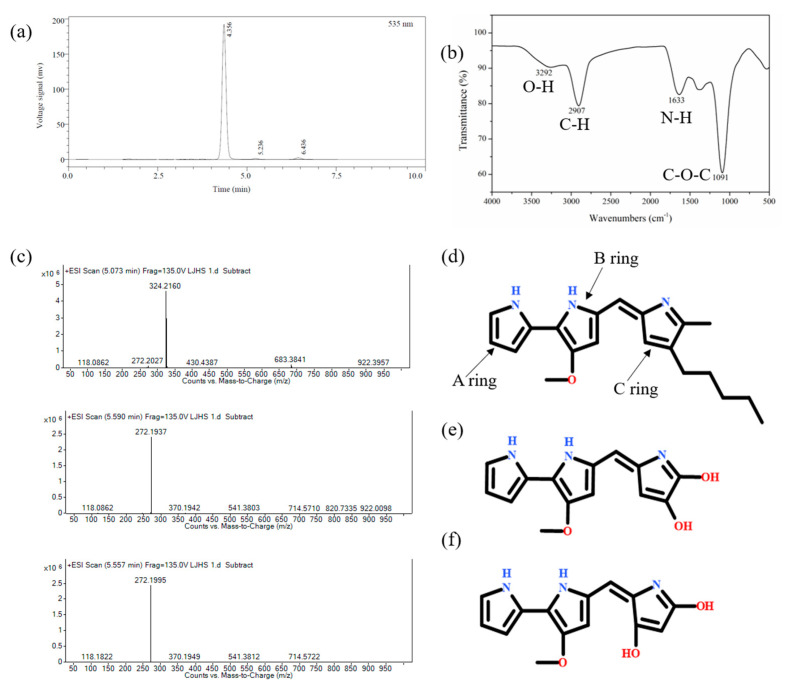
Spectral analysis of the red pigment in the fermentation broth. (**a**) Infrared spectrum, (**b**,**c**) HPLC-MS of crude prodigiosin. (**d**–**f**) Structure prediction of the metabolites based on HPLC-MS. Structure of the main product (**d**) and the two by-products (**e**,**f**).

**Figure 4 molecules-26-02156-f004:**
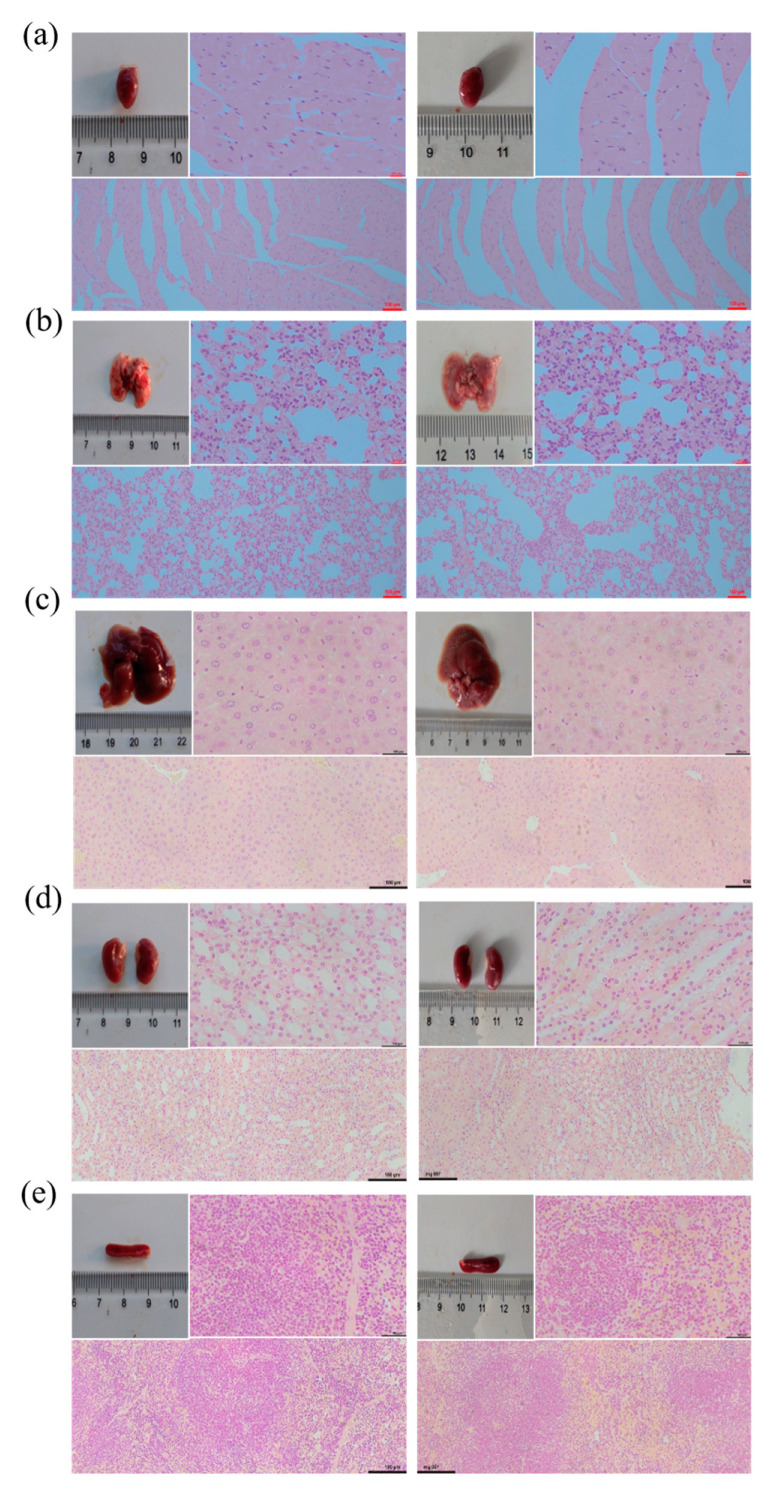
Effect of prodigiosin on the appearance of the heart (**a**), liver (**b**), spleen (**c**), lung (**d**), kidney (**e**) tissues of mice (left images represent organs and tissues from the group prodigiosin group (PF), right images represent organs and tissues from the C group).

**Figure 5 molecules-26-02156-f005:**
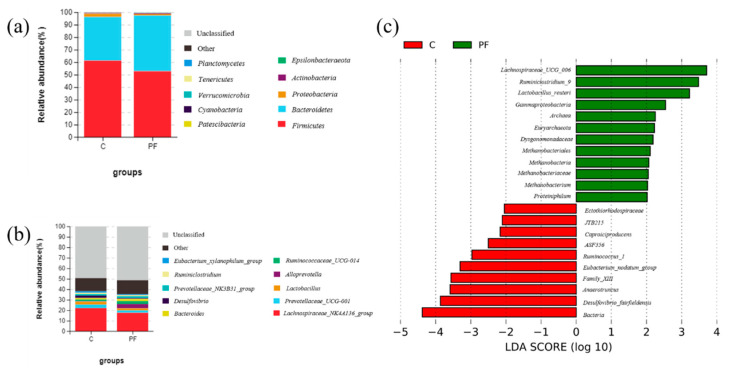
Relative abundance distribution of the 10 most abundant microbial taxa at the phylum level (**a**) and genus level (**b**). (**c**) LEfSe score plot of the discriminative microbial taxa (LDA score > 2) that are highly enriched in the C and PF groups.

**Figure 6 molecules-26-02156-f006:**
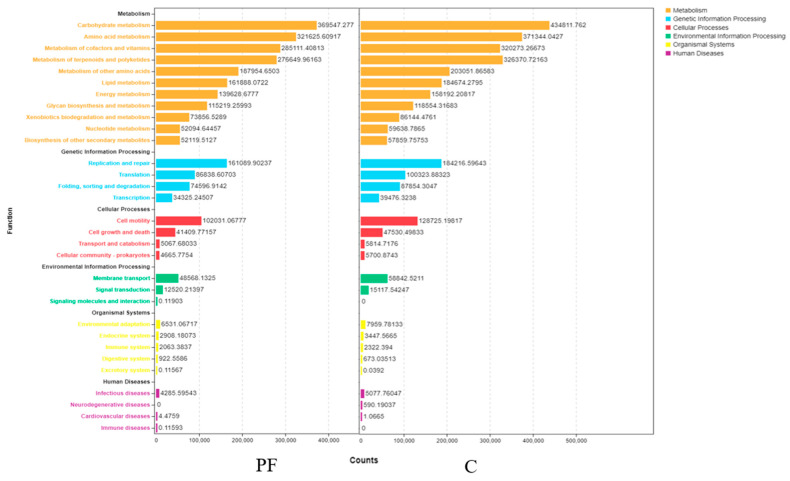
Function distribution stack based on PICRUSt2 of group C (**right**) and group PF (**left**).

**Table 1 molecules-26-02156-t001:** Bodyweight changes in mice treated with prodigiosin for 18 days.

BW/g	C	PF
d1	44.498 ± 1.5	45.786 ± 2.0
d6	44.704 ± 2.0	45.726 ± 1.2
d12	45.242 ± 1.7	46.926 ± 1.5
d18	46.126 ± 1.4	46.944 ± 0.9

## Data Availability

Nucleotide sequences of 16S rRNA gene of the strain ZPG19 isolated from the compost samples were deposited in GenBank under the accession numbers MW308502. Sequence data are available at NCBI under accession ID PRJNA670576. All strains are available upon request from the corresponding author.

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
