# Peer review of "Prodigiosin of Serratia marcescens ZPG19 Alters the Gut Microbiota Composition of Kunming Mice"

_molecules, 2021, doi:10.3390/molecules26082156_

Round 1

Reviewer 1 Report

This manuscript reports on the isolation and identification of prodigiosin, a red pigment synthesized by Serratia marcescens. This metabolite has been previously studied for its biological activities and broadly used in several fields including ecosystem management, pharmaceutical, and cosmetics. However, this manuscript is the first to characterize and assess the impact of prodigiosin on gut microbiome composition.

Overall, the manuscript is very well organized and written. It contains interesting findings that might be of great interest for Molecules readers especially those working on microbiomes’ regulation.  Some minor revisions are requested before acceptance for publication in Molecules. Here are some specific edits that should be done to improve the current version of the manuscript:

  • The introduction is well-written and emphasizes the prodigiosin’s description, classification, and applications. However, I suggest adding a small paragraph to state the previous use and importance of bacterial metabolites in microbiota regulation.
  • In paragraph 2.3 the authors mentioned that they verify the identity of prodigiosin using HPLC-MS. Did they look at fragmentation patterns and compared it to existing literature? Why they did not use a commercial standard that is available at Sigma or others?
  • In paragraph 2.6, please precise what Statistical Analysis Software have you used.
  • In the discussion, the authors start several sentences with an abbreviation. You should never use abbreviations at the beginning of the sentence (examples: reuteri, D. fairfieldensis, etc.). Please modify throughout the manuscript.
  • In the discussion as well, the reviewer suggests citing and discussing papers that were published on similar research to emphasize on the application nowadays of microbial compounds for microbiomes regulation.

Reviewer 2 Report

INTRODUCTION

Please provide structures of the compounds

METHODS

Please convert centrifugation speeds from rpm to rcf (x g) throughout the manuscript

Please provide more specific conditions for the analytical characterization of the pigment

It is hard to believe that a crude methanolic extract from a microbial culture would yield 95% pure prodigiosin. How was this purity determined?

Please provide the name of the animal use approval body and the approved protocol number

How was the dose (150 μg/kg/d) of prodigiosin selected?

Please be more specific of the dosing conditions. What were the concentrations of prodigiosin and ethanol in both the stock prodigiosin solution and in the final drinking water?

Why were 5 mice/group used for the experiment but only 3 mice/group used for analysis?

RESULTS

The mice were 28 g when purchased by 45 g when used for the experiment…what happened to them in between?

In section 3.6, the authors refer to pathways not shown on Figure 5

DISCUSSION

Based on Figures 4-5, the changes in the microbiome were very modest

Little here to suggest that it is a promising candidate drug…no disease states were evaluated. These were healthy mice
